# miR-195-5p as Regulator of γ-Catenin and Desmosome Junctions in Colorectal Cancer

**DOI:** 10.3390/ijms242317084

**Published:** 2023-12-03

**Authors:** Emanuele Piccinno, Viviana Scalavino, Raffaele Armentano, Gianluigi Giannelli, Grazia Serino

**Affiliations:** National Institute of Gastroenterology S. De Bellis, IRCCS Research Hospital, Via Turi 27, 70013 Castellana Grotte, BA, Italy; emanuele.piccinno@irccsdebellis.it (E.P.); viviana.scalavino@irccsdebellis.it (V.S.); raffaele.armentano@irccsdebellis.it (R.A.); gianluigi.giannelli@irccsdebellis.it (G.G.)

**Keywords:** microRNA, CRC, JUP, γ-catenin, miR-195-5p, desmosome, cell junctions

## Abstract

Desmosomes play a key role in the regulation of cell adhesion and signaling. Dysregulation of the desmosome complex is associated with the loss of epithelial cell polarity and disorganized tissue architecture typical of colorectal cancer (CRC). The aim of this study was to investigate and characterize the effect of miR-195-5p on desmosomal junction regulation in CRC. In detail, we proposed to investigate the deregulation of miR-195-5p and *JUP*, a gene target that encodes a desmosome component in CRC patients. JUP closely interacts with desmosomal cadherins, and downstream, it regulates several intracellular transduction factors. We restored the miR-195-5p levels by transient transfection in colonic epithelial cells to examine the effects of miR-195-5p on JUP mRNA and protein expression. The JUP regulation by miR-195-5p, in turn, determined a modulation of desmosome cadherins (Desmoglein 2 and Desmocollin 2). Furthermore, we focused on whether the miR-195-5p gain of function was also able to modulate the expression of key components of Wnt signaling, such as NLK, LEF1 and Cyclin D1. In conclusion, we have identified a novel mechanism controlled by miR-195-5p in the regulation of adhesive junctions, suggesting its potential clinical relevance for future miRNA-based therapy in CRC.

## 1. Introduction

Colorectal cancer (CRC) is a multifactorial disorder involving genetic and environmental factors that affect intestinal homeostasis. Several studies have suggested a link between chronic inflammation and CRC, in which gastrointestinal inflammation may contribute to the onset of CRC [1,2,3]. The incidence of CRC in IBD (inflammatory bowel disease) patients (i.e., colitis-associated cancer or CAC) was reported to be higher [4], and in sporadic CRCs, there is a prominent inflammatory response known as tumor-induced inflammation. In this context, the loss of intestinal epithelial barrier function after inflammatory stimuli is recognized as a key event in the pathogenesis of IBD, which may evolve into colorectal cancer [5].

Tight junctions (TJs), adherens junctions (AJs) and desmosomes are the three components of the junctional complex responsible for the paracellular space between epithelial cells. TJs and AJs are the best-characterized junctional complexes within the gut. Furthermore, desmosomal adhesion and associated intracellular signaling play a significant role in maintaining the intestinal barrier in both healthy and diseased people [6]. 

Desmosome complexes contribute to the establishment of apical–basal polarity in epithelial cells, which markedly changes during colorectal tumor progression. The dysregulation of desmosome-associated proteins is associated with a loss of epithelial cell polarity and disorganized tissue architecture [6]. 

JUP, also known as Junction Plakoglobin or γ-catenin, is a member of the Armadillo family of proteins, which is structurally and functionally a homolog of β-catenin. JUP interacts with desmosomal cadherins, desmoglein and desmocollin, promoting intermediate filament associations with the membrane and conferring tensile strength and resilience to cells. In addition to its cell–cell adhesive functions, JUP is involved in intracellular signaling, interacting with a number of signaling proteins and transcription factors [7].

The highly conserved central regions of γ- and ß-catenin facilitate their binding to various proteins, including the adenomatous polyposis coli (APC) tumor suppressor, T-cell factor/lymphoid enhancer factor (Tcf/Lef) transcription factors and axin/conductin proteins [8,9,10]. γ-catenin can bind APC, which modulates its cytoplasmic and nuclear levels with axin and GSK3 [11,12,13]. Following Wnt activation, JUP accumulates in the cytoplasm and nucleus, where it binds members of the Tcf/Lef transcription factor family, increasing cyclin D1, c-myc and MMP-7 expression [14]. These findings suggest that JUP also has an oncogenic role. 

MicroRNAs (miRNAs) are a family of small (21–23 nt) non-coding single-stranded, endogenous RNA molecules. They post-transcriptionally regulate gene expression by either mediating translational repression or directing mRNA cleavage, mainly binding complementary sequences in the 3′UTR of mRNA [15]. 

Several studies have also analyzed the correlation between miRNAs and many diseases [16,17,18,19,20,21,22], including CRC [23,24,25,26,27]. Much evidence suggests a potential clinical role of miRNAs in CRC progression via tight [28,29] and adherence junction [30,31] regulation. Ke and co-workers revealed that miR-103 had strong tumor-promoting effects targeting ZO-1 in CRC [29]. Gulei and colleagues identified a downregulated profile for miR-205-5p in colon adenocarcinoma patients and demonstrated that the exogenous upregulation of this miRNA is able to significantly raise the levels of E-cadherin through the direct inhibition of ZEB1 [32]. Zhang and colleagues found that miR-155 is significantly upregulated in CRC patients and plays an important role in promoting CRC progression through the regulation of claudin-1 expression [33]. It has also been demonstrated that the inhibition of miR-200c correlated with the acquired resistance of colorectal cancer cells (HCT-116) to 5-FU, with decreasing levels of E-cadherin and PTEN protein and with tumor growth [34]. Furthermore, in our recent papers, we demonstrated that miR-195-5p modulates CLDN2 and other components of TJs, highlighting the role of this miRNA in ensuring barrier integrity [35,36]. 

The involvement of miRNAs in desmosome complex regulation is still poorly investigated. The aim of this study was to characterize the role of miR-195-5p on desmosomal junction regulation and its potential effect on CRC development. Specifically, we propose to report the miR-195-5p and *JUP* deregulation in CRC patients and to investigate whether the restoration of miR-195-5p levels regulates JUP expression and, in turn, the expression of the desmosomal cadherins, DSG2 and DSC2, closely associated with JUP. In addition, we aimed to study whether JUP regulation by miR-195-5p could influence the expression of key effectors of the Wnt pathway, such as NLK, LEF1 and Cyclin D1.

## 2. Results

### 2.1. miR-195-5p Expression in CRC Patients

To assess whether miR-195-5p expression was dysregulated in colorectal cancer compared to healthy tissue, we analyzed a GEO dataset (https://www.ncbi.nlm.nih.gov/geo/query/acc.cgi?acc=GSE115513; GSE115513; accessed on 20 January 2023) [37] that includes miRNA expression profiles of patients with CRC and adjacent normal tissue. The obtained data, expressed as the mean of 411 CRC tissues and 381 adjacent normal samples, highlighted a significant downregulation of miR-195-5p in CRC samples (*p* < 0.0001; Figure 1).

### 2.2. miR-195-5p as a Potential Modulator of JUP

To study the molecular mechanism of colorectal cancer involving miR-195-5p, we performed a bioinformatic analysis to predict its target genes. The results obtained with the miRabel database underlined that *JUP* is one putative target of miR-195-5p (Appendix A in the Appendix A). This gene encodes for a desmosome protein physiologically expressed in colonic epithelium, which is essential to guarantee intestinal homeostasis.

### 2.3. JUP Expression in CRC Patients

To compare *JUP* expression between colorectal cancer and adjacent normal colon, we used the GEPIA tool, an interactive web application for gene expression analysis of tumors and normal samples from the TCGA and the GTEx databases [38]. Our analysis showed that *JUP* was significantly upregulated in CRC patients compared with healthy controls (*p* < 0.05; Figure 2).

miR-195-5p and *JUP* expression in CRC were also further evaluated by analyzing a GEO dataset that includes both the miRNA and the mRNA expression profile in the same cohort of CRC patients [39]. As shown in Figure 3A,B, the miR-195-5p and *JUP* levels were markedly deregulated in CRC tissues compared with adjacent normal colon (*p* < 0.001). Moreover, Pearson correlation analysis showed that their expression was inversely correlated (Figure 3C, r = −0.689, R^2^ = 0.475, *p* < 0.0001), suggesting that *JUP* may be regulated by miR-195-5p.

### 2.4. miR-195-5p Is an Effective Regulator of JUP mRNA

Our analysis revealed that miR-195-5p was strongly downregulated in CRC patients, while one of its putative targets, *JUP*, was significantly overexpressed. These results suggest that miR-195-5p dysregulation could be the cause of *JUP* deregulation and affected desmosome function. To functionally validate this hypothesis, we carried out in vitro transient transfection with synthetic molecules of miR-195-5p mimic in Caco2 and LoVo cell lines. The effect of miR-195-5p upregulation on its candidate gene target, *JUP*, was investigated by real-time PCR. We evaluated the miR-195-5p expression upon mimic transfection, and our results highlighted a significant increase of miR195-5p levels in both cell lines (*p* < 0.05; Figure 4A). Moreover, the transient transfection with miR-195-5p mimic at the 30 nM and 50 nM concentrations significantly decreased *JUP* mRNA expression in both cell lines (*p* < 0.001; Figure 4B).

### 2.5. miR-195-5p Directly Regulates γ-Catenin Protein Expression 

To further characterize the role of miR-195-5p in the regulation of γ-catenin protein expression, we performed Western blot analysis. Enhancing the mature form of miR-195-5p in Caco2 and LoVo cell lines, we evaluated γ-catenin protein levels. In accordance with the RNA results, we found that γ-catenin protein expression decreased after transient transfection in Caco2 (*p* < 0.001; Figure 5A) and LoVo (*p* < 0.001; Figure 5A) cell lines compared with mock control.

Furthermore, immunofluorescence staining in the Caco2 and LoVo monolayers additionally demonstrated that miR-195-5p mimic transfection at 30 nM and 50 nM significantly reduced γ-catenin protein compared to mock control (Figure 5B). The intensity of γ-catenin positivity was quantified and expressed as a function of the number of cells. The mean quantification of immunofluorescence images confirmed once more that the γ-catenin signal intensity was significantly downregulated after miR-195-5p mimic transfection (*p* < 0.0001; Appendix A in the Appendix A)

### 2.6. miR-195-5p Indirectly Modulates the Desmosome Cadherins DSG2 and DSC2

To analyze the potential effect of the miR-195-5p increase on adhesive junctions of epithelial cells, we also evaluated the expression levels of other structural desmosomal proteins that are closely associated with γ-catenin. Desmoglein-2 (DSG2) and Desmocollin-2 (DSC2) are transmembrane proteins that regulate intercellular connections, contributing to desmosome assembly and playing an important role in colon cancer progression. 

Our experimental data demonstrated that after raising miR-195-5p levels to 30 nM and 50 nM concentrations, DSG2 and DSC2 proteins resulted upregulated in all cell lines (*p* < 0.01; Figure 6A,B). These results revealed that miR-195-5p was able to indirectly modulate the expression of desmosome cadherins via γ-catenin. 

### 2.7. miR-195-5p Regulates Wnt Pathway Activation 

JUP, as a structurally and functionally homologous variant of β-catenin, can trigger the Wnt pathway and may lead to the aberrant activation of its canonic signaling. JUP accumulation mediates the transcriptional induction of target genes involved in cell proliferation and cancer progression [40,41,42,43,44,45]. To investigate the effect of miR-195-5p on Wnt intracellular signal transduction, we investigated the protein expression of other key effectors of this pathway. Interestingly, our results showed that miR-195-5p induction significantly decreased the protein levels of NLK, LEF-1 and Cyclin D1 in Caco2 (*p* < 0.001; Figure 7A) and LoVo cells (*p* < 0.001; Figure 7B). These main components of the Wnt pathway were remarkably reduced in transfected conditions at both the 30 and 50 nM mimic concentrations.

### 2.8. miR-195-5p Inhibits Cell Migration in CRC

To further study the involvement of miR-195-5p in the regulation of desmosome function, a transwell migration assay was carried out. The gain of miR-195-5p in Caco2 and LoVo cell lines strongly reduced the migration potential and activity of cells as compared with mock control. These data demonstrated that the transfection with miR-195-5p mimic at 30 nM and 50 nM concentration reduced the migration ability of colon cancer cells in vitro (*** *p* < 0.0001; Figure 8).

## 3. Discussion

Desmosomes mediate the intercellular adhesion to intermediate filament cytoskeletons and affect tight and adherence junctions, conferring intestinal epithelial integrity and homeostasis. Altered cell junction expression is directly associated with the loss of intestinal barrier function and leads to the onset of intestinal disorders. In our recent previous works, we underlined the role of miR-195-5p in regulating claudin-2 (CLDN2), a tight junction aberrantly expressed in the disrupted colonic epithelial barrier of patients with IBD. We demonstrated that the abnormal expression of CLDN2 in IBD patients was due to miR-195-5p downregulation, and an increased expression in colonic epithelial cells was able to restore normal levels of this protein [35]. Subsequently, in vitro and in vivo, we verified that miR-195-5p was able to ameliorate the colonic inflammatory response [36]. Moreover, it has been demonstrated that the overexpression of miR-122 caused by pathologic inflamed conditions induced occludin mRNA degradation. The occludin depletion from enterocytes increased the intestinal tight junction permeability in Caco-2 monolayers as well as in mouse intestine intestinal model systems [46]. Similarly, it has been proved that miR-874 can suppress occludin and claudin-1 expression, leading to a substantial enhancement of the paracellular permeability in vitro through targeting 3′ UTR of AQP3 [47].

The abnormal expression of miRNAs in cancer was also widely investigated. It was demonstrated that miR-514b-5p acts as an oncogene in CRC tumorigenesis targeting Claudin1 (CLDN1) and E-cadherin (CDH1) [48]. Tang and colleagues have found that the increased expression of miR-29a promoted CRC progression by regulating CDH1/MMP2 through direct targeting of KLF4. Specifically, the authors showed that miR-29a promoted MMP2 and decreased E-cad expression, highlighting the crucial regulation of cell junctions by miRNAs [49]. 

Although these findings further underline the role of miRNAs in the regulation of cell junctions and their key action in cancer progression, to the best of our knowledge, there are no available studies describing the relationship between miRNAs and desmosome regulation in CRC.

Desmosome intercellular junctions are essential not only to ensure the integrity of the intestinal epithelium but also to the mechanical machinery necessary to execute complex morphogenetic and homeostatic intercellular rearrangements. They act as signaling hubs that integrate mechanical and chemical pathways to coordinate the tissue architecture. JUP is a closely related homolog of ß-catenin and acts in cell adhesion as a key component of the desmosome complex, but many studies have reported its involvement in cell signaling [12,13,14]. However, JUP functions as a transcriptional factor have not been revealed in detail. Some findings suggest that both JUP and ß-catenin share common protein partners and may perform some of the same functions. JUP expression, similar to that of ß-catenin, is regulated by Wnt signaling through axin/APC complex-induced proteasomal degradation. Maeda and colleagues showed, by immunoprecipitation, that JUP binds to TCF/LEF, and by performing a luciferase reporter assay, that it has a TCF/LEF-dependent transcriptional activity [50]. The JUP contribution to the activation of Tcf/Lef transcription was further assessed, improving our knowledge of its involvement in the Wnt pathway [44,45,51,52].

In this study, using a public data repository, we reported that miR-195-5p was downregulated in colon cancer tissues; inversely, *JUP* mRNA expression was found to be increased in CRC patients compared with normal counterpart tissues. This opposite expression suggests that *JUP* could be one of the gene targets of miR-195-5p. Using the strategy to gain the function of miR-195-5p, we biologically confirmed that JUP mRNA and protein expression were strictly regulated by this miRNA. This modulation by miR-195-5p promoted, in turn, an indirect up-regulation of the desmosome cadherins, DSG2 and DSC2, at the protein level. These results support the potential relevance of miR-195-5p in the regulation of adhesive junctions and its key role in cancer development. Moreover, since JUP was reported to be involved as a transcription factor in cell signaling, we investigated whether miR-195-5p also modulated the expression of other proteins implicated in the Wnt pathway. Interestingly, in colon cancer cell lines, we found that increasing intracellular miR-195-5p was able to reduce the protein expression levels of NLK, LEF1 and Cyclin D1 via JUP modulation. 

In addition, we have further investigated the effects of miR-195-5p in the regulation of desmosome junctions by performing a functional biological assay. The transwell assay revealed that the increase in intracellular levels of miR-195-5p significantly inhibits the migration ability of CRC cell lines. These findings are strongly in accordance with the results obtained by Western blot analysis, which have shown an indirect upregulation of desmosomal cadherins by miR-195-5p and suggest a protective role of miR-195-5p in the establishment of cell–cell adhesions.

Altogether, these results highlighted, for the first time, an interesting mechanism controlled by miR-195-5p in the regulation of adhesive junction in CRC and provided candidate targets for CRC treatment. However, this was a preliminary study aimed at biologically investigating the involvement of miR-195-5p in CRC and in desmosome complex regulation. Future studies will be needed to validate the positive effectiveness of miR-195-5p in CRC progression.

## 4. Materials and Methods

### 4.1. Cell Cultures

For all experiments, two human colon cell lines were used. We chose to work with the Caco2 and LoVo cell lines because these two models allow full characterization of the epithelial junctions. In fact, below, we report several features for each cell line. Caco2 and LoVo were purchased from ATCC (American Type Culture Collection, Manassas, VA, USA). The cell lines were grown in culture medium composed of Dulbecco’s Modified Eagle Medium (DMEM, Thermo Fisher Scientific, Waltham, MA, USA) with heat-inactivated Fetal Bovine Serum (10% for Caco2 and 20% for LoVo) (FBS, Thermo Fisher Scientific, Waltham, MA, USA), 10 mM HEPES (Sigma-Aldrich, St. Louis, MO, USA), 1 mM sodium pyruvate (Sigma-Aldrich, St. Louis, MO, USA) and 1% streptomycin/penicillin (Thermo Fisher Scientific, Waltham, MA, USA).

### 4.2. In Vitro Transfection

All cell lines were seeded in 12-well and 6-well plates for RNA and protein extraction (Corning, Corning, NY, USA), respectively, and in Lab-Tek Chamber Slides for immunofluorescence assays. 

Caco2 and LoVo cell lines were transfected with synthetic molecules of miR-195-5p mimic at concentrations of 30 nM and 50 nM (Life Technologies, Carlsbad, CA, USA) using TKO transfection reagent (Mirus Bio LLC, Madison, WI, USA), according to the manufacturer’s instructions. 

In all transfection experiments, we included a mock control in which cells were transfected with transfection reagent but without miRNA mimic.

### 4.3. RNA Extraction and Real-Time PCR 

Twenty-four hours after transfection, total RNA was extracted from cell cultures using TRIzol reagent (Invitrogen by Thermo Fisher Scientific, Waltham, MA, USA) according to the manufacturer’s recommendations.

Total RNA was eluted in ribonuclease-free water, and the concentrations were determined with the NanoDrop ND-2000 Spectrophotometer (Thermo Fisher Scientific, Waltham, MA, USA).

To analyze the miR-195-5p expression upon transfections, total RNA was reverse transcribed using a TaqMan Advanced miRNA cDNA Synthesis Kit (Thermo Fisher Scientific, MA, USA), according to protocol. For miR-195-5p quantification, the RT-PCR was performed using 20 μL of final volume on a CFX96 System (Biorad Laboratories, CA, USA) with TaqMan Advanced miRNA assays and TaqMan Fast Advanced Master mix (Thermo Fisher Scientific, MA, USA). Data were normalized using miR-26a-5p as endogenous control. The relative expression was calculated using the 2^−ΔCt^ formula.

For *JUP* detection, cDNA was prepared with the iScript Reverse Transcription Supermix (BioRad Laboratories, CA, USA) following the manufacturer’s protocol. Quantitative real-time PCR was performed on a CFX96 System (Biorad Laboratories, Hercules, CA, USA) using the SsoAdvanced Universal SYBR Green Supermix (BioRad Laboratories, Hercules, CA, USA) and the QuantiTect Primer Assay for *JUP* and *GAPDH* (Qiagen, Hilden, Germany). Comparative qPCR was carried out in triplicate, and *GAPDH* gene amplification was used as reference standard to normalize the relative expression of *JUP*. The relative expression was calculated using the 2^−ΔΔCt^ formula.

### 4.4. Western Blot

Total protein extraction was obtained using T-PER Tissue Protein Extraction Reagent (Thermo Fisher Scientific, Waltham, MA, USA) supplemented with cocktail proteinase inhibitors (Sigma-Aldrich, St. Louis, MO, USA) seventy-two hours after transfection.

Total protein concentration was determined with the Bradford colorimetric assay (Bio-Rad Laboratories, Richmond, CA, USA) and an equal amount of proteins were separated on 4–20% Mini-PROTEAN TGX Stain-Free Protein Gels (Biorad Laboratories, Hercules, CA, USA) and then electrotransferred on 0.2 μm pore size PVDF membranes (Biorad Laboratories, Hercules, CA, USA). For analysis, PVDFs were incubated in an automated iBind Flex Western Device (Thermo Fisher Scientific, Waltham, MA, USA) with primary and secondary antibodies, and the signal intensities were detected using the Chemidoc System (Biorad Laboratories, Hercules, CA, USA). Images were analyzed with Image Lab Software version 5.2.1 (Biorad Laboratories, Hercules, CA, USA) and quantified by ImageJ Software 1.54d. β-Tubulin protein was used as housekeeping value to normalize the target protein signal.

For protein detection, primary antibodies of rabbit monoclonal γ-Catenin (#75550S, Cell Signaling, Technology, Danvers, MA, USA; dilution 1:3000), rabbit polyclonal Desmoglein-2 (ab226184, Abcam, Cambridge, UK; dilution 1:5000), mouse monoclonal Desmocollin-2 (#32-6200, Invitrogen, Carlsbad, CA, USA; dilution 1:500) rabbit polyclonal NLK (#PA5-21877, Invitrogen, Carlsbad, CA, USA; dilution 1:500), mouse polyclonal LEF-1 (#MA1-12420, Invitrogen, Carlsbad, CA, USA; dilution 1:400), mouse monoclonal Cyclin D1 (#MA5-16356, Invitrogen, Carlsbad, CA, USA; dilution 1:1000) and mouse monoclonal ß-Tubulin (sc-166729, Santa Cruz Biotechnology, Inc., Heidelberg, Germany; dilution 1:1000) were used.

Secondary antibodies included Goat Anti-mouse IgG-(H+L)-HRP conjugate (170-6516, Biorad Laboratories, CA, USA; dilution 1:500) and Goat Anti-rabbit IgG-(H+L)-HRP conjugate (#31466, Invitrogen, Carlsbad, CA, USA; dilution 1:2500).

### 4.5. Immunofluorescence

Caco2 and LoVo cell lines were seeded in Lab-Tek Chamber Slides and transfected when confluence was reached. After transfection, cells were washed with PBS and fixed with PFA 4% for 10 min at 4 °C and subsequently washed twice with PBS. Permeabilization was performed by incubating the monolayer with Triton-X 0.1% in PBS for 5 min at room temperature. Samples were blocked in PBS + BSA 3% for 1.5 h at room temperature. Then, samples were stained with primary antibody rabbit polyclonal γ-Catenin (#75550S, Cell Signaling, Technology, Danvers, MA, USA; dilution 1:100) and diluted in PBS + BSA 3% for 3 h. After washing with PBS, they were stained with secondary antibody chicken anti-Rabbit IgG (H + L) Alexa Fluor 594 (A-21442, Invitrogen, Carlsbad, CA, USA, dilution 1:400) in PBS +BSA 3% for 1 h. ProLong Gold Antifade Mountant with DAPI (Thermo Fisher Scientific, Waltham, MA, USA) was applied to each sample then mounted with a glass coverslip. The fluorescence was observed using an Eclipse Ti2 Nikon microscope (Nikon Inc., Melville, NY, USA). Four images were captured in different positions for each sample.

### 4.6. Migration Assay

For the migration assay, Caco2 and LoVo were initially seeded and transfected in 6 wells. Subsequently, cells were reseeded in serum-free media into the upper chamber of the 6.5 mm Transwell^®^ inserts (8 µm pore size; Corning, Corning, NY, USA), coated with Collagen I. In the lower chamber, complete medium supplemented with 10% FBS was added. After 24 h, cells that migrated were fixed using 4% paraformaldehyde and stained with 0.1% crystal violet (Sigma-Aldrich, St. Louis, MO, USA). The fixed cells were counted by light microscope in randomly selected fields.

### 4.7. Bioinformatic and Statistical Analyses 

Differential miR-195-5p expression analysis in CRC samples compared to the respective normal tissue was assessed based on the Gene Expression Omnibus database (GEO, https://www.ncbi.nlm.nih.gov/geo; GSE115513; accessed on 20 January 2023; GSE126093 accessed on 20 November 2023) [37,39]. 

The GEPIA tool (http://gepia.cancer-pku.cn/index.html, accessed on 6 March 2023 [38]) and GSE126092 (accessed on 20 November 2023) [39] were used to perform a differential JUP expression analysis between colorectal cancer and adjacent normal tissue. 

To assess putative miRNA gene targets, we applied the miRabel algorithm (http://bioinfo.univ-rouen.fr/mirabel/; accessed on 15 February 2023) [53]. 

Statistical analysis was performed using GraphPad Prism software version 9.0.0. Statistical significance was evaluated with two-tailed Student’s *t*-test. A Pearson correlation test was applied to study continuous variables. All values are expressed as the mean ± SEM of data obtained from at least three independent experiments. Results were considered statistically significant at *p* < 0.05.

## 5. Conclusions

In conclusion, our results show, for the first time, that an intracellular increase in miR-195-5p, which is downregulated in CRC, can reverse the higher levels of JUP. Indeed, an increased expression of JUP is revealed to be strongly associated with the onset of colorectal cancer. miR-195-5p was indirectly able to regulate the desmosome cadherins and other key components of the Wnt pathway.

## Figures and Tables

**Figure 1 ijms-24-17084-f001:**
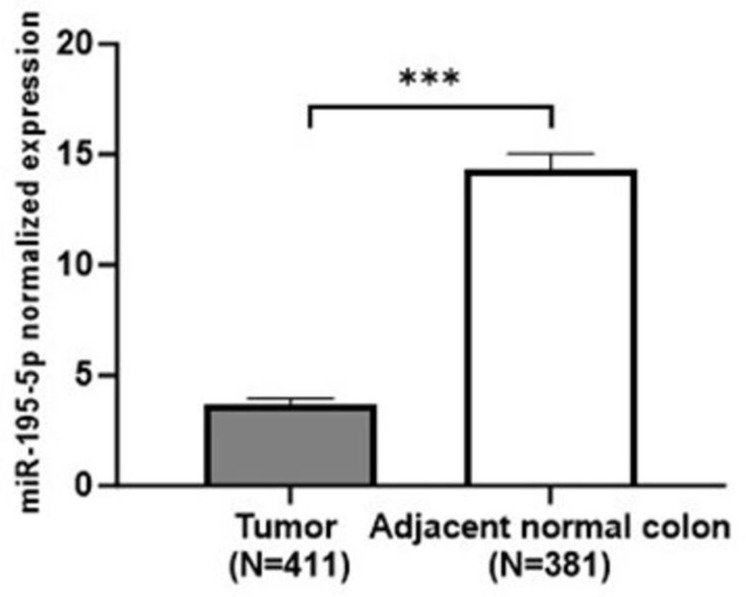
miR-195-5p expression in tumor and adjacent normal colon. Differential miRNA expression analysis performed based on the GEO database (GSE115513) revealed that miR-195-5p was significantly decreased in CRC tissue compared with adjacent normal samples. *** *p* < 0.0001.

**Figure 2 ijms-24-17084-f002:**
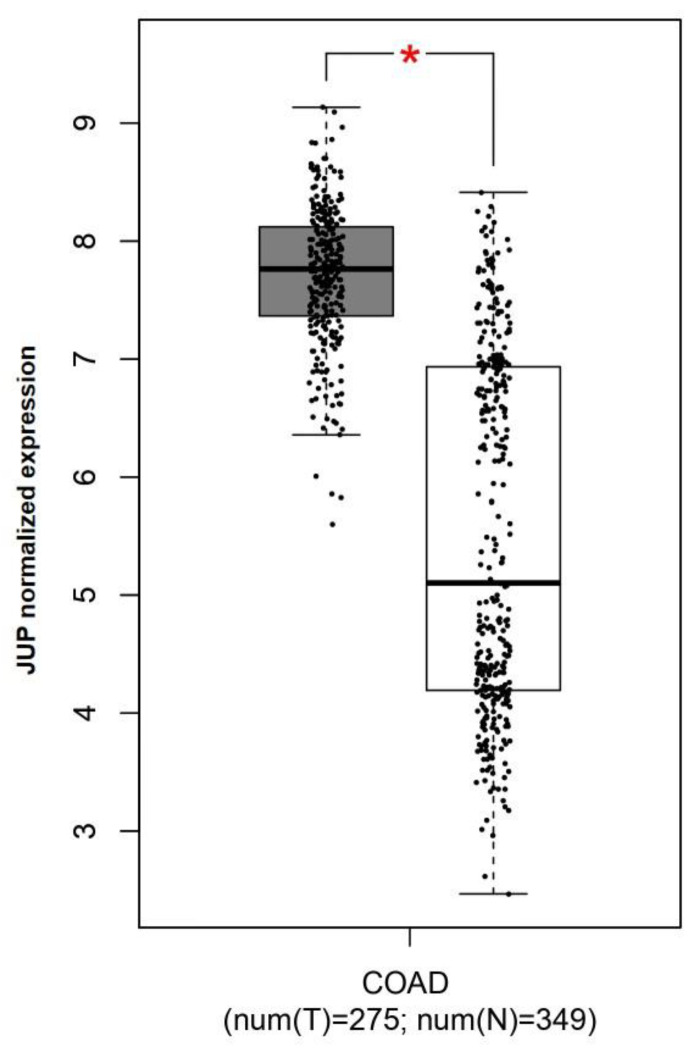
*JUP* mRNA expression in CRC and adjacent normal colon tissues. The results obtained from investigation with the GEPIA tool are reported as the mean of 275 tumor and 349 normal samples, demonstrating a statistically significant higher expression in tumor. * *p* < 0.05.

**Figure 3 ijms-24-17084-f003:**
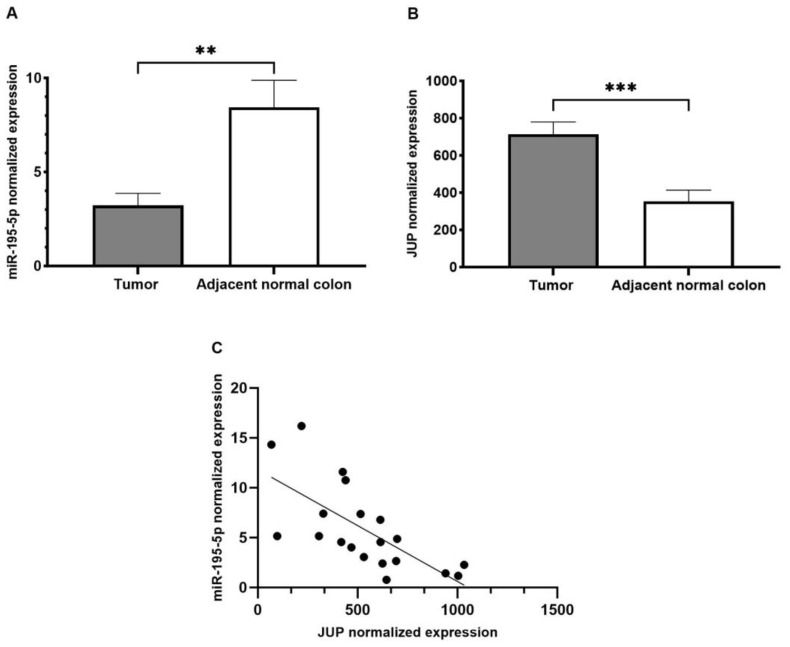
miR-195-5p and *JUP* expression in CRC patients. Data from GEO dataset (GSE126092, GSE126093), which includes the miRNA and mRNA expression profile of 10 CRC patients with tumor tissue and the relative adjacent normal samples, demonstrated a considerable downregulation of miR-195 in tumor (**A**). Inversely, *JUP* was significantly overexpressed in CRC tissue compared with adjacent normal colon (**B**). ** *p* < 0.001, *** *p* < 0.0001. Pearson correlation analysis in all samples revealed a negative correlation between miR-195-5p and *JUP* expression. r = −0.689, R^2^ = 0.475, *p* < 0.0001. Each point represents a sample (**C**).

**Figure 4 ijms-24-17084-f004:**
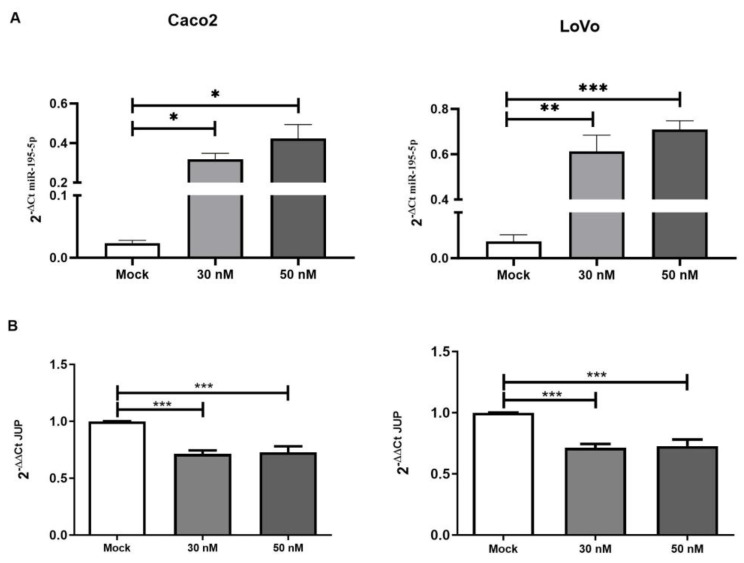
miR-195-5p (**A**) and *JUP* expression (**B**) in Caco2 and Lovo cell lines after transient transfection. miR-195-5p was markedly overexpressed at 30 nM and 50 nM concentrations compared with mock control. The gain of function of miR-195-5p regulates *JUP* mRNA expression. The increased intracellular miR-195-5p at 30 nM and 50 nM concentrations led to a significant decrease of *JUP* in all cell lines. Expression data were normalized to the housekeeping gene *GAPDH* and are representative of four independent experiments (n = 4; mean ± SEM) * *p* < 0.05, ** *p* < 0.001, *** *p* < 0.0001.

**Figure 5 ijms-24-17084-f005:**
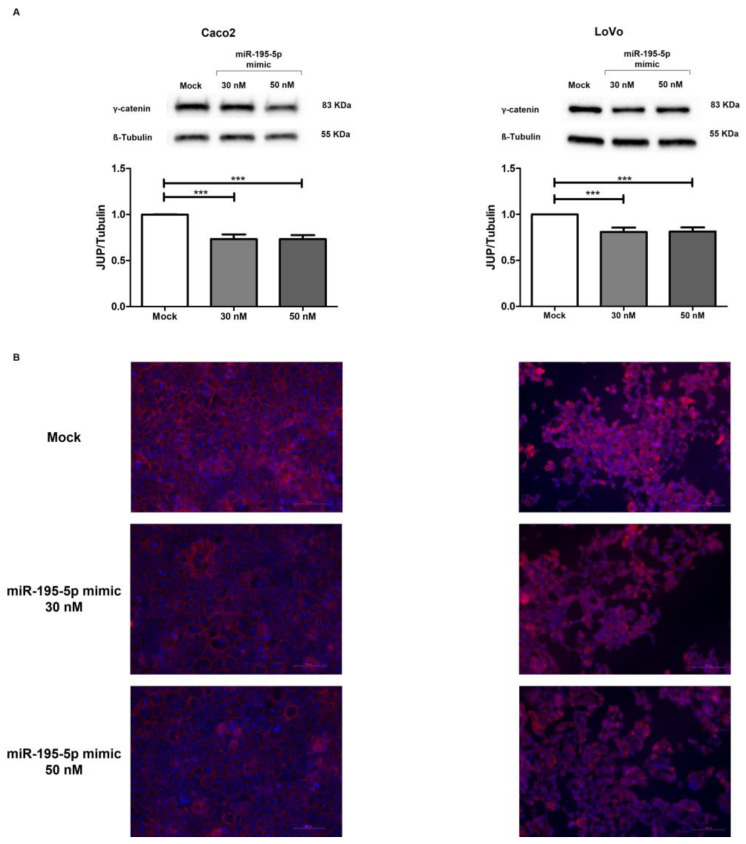
Regulation of γ-catenin protein expression by miR-195-5p in colonic epithelial cell lines. (**A**) Western blot analysis of γ-catenin protein expression after miR-195-5p mimic transfection. A significant reduction of γ-catenin expression was detected in all cell lines compared with mock control. WB data were obtained by dividing the normalized transfected sample values by the normalized control sample values. β-Tubulin was used as housekeeping protein to normalize the data obtained in four independent experiments (n = 4; mean ± SEM); *** *p* < 0.0001. (**B**) Immunofluorescence staining of γ-catenin in Caco2 and LoVo monolayers transfected with miR-195-5p mimic at 30 nM and 50 nM. In accordance with Western blot results, after transfection, the expression of γ-catenin is greatly reduced. The images were acquired at 20× magnification. Single-channel images of DAPI and γ-Catenin staining are merged. Scale bar 50 μm.

**Figure 6 ijms-24-17084-f006:**
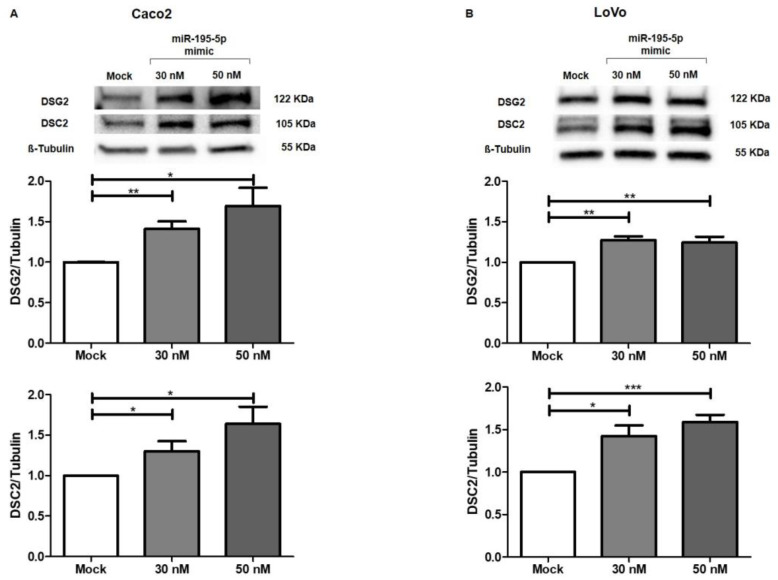
Regulation of desmosome-associated protein expression by miR-195-5p mimic in Caco2 (**A**) and Lovo (**B**) cell lines. The transient transfection with miR-195-5p at 30 nM and 50 nM concentrations led to a significant decrease of γ-catenin protein expression that, in turn, indirectly modulated DSG2 and DSC2. Data are representative of four independent experiments (n = 4; mean ± SEM) and were normalized to β-Tubulin housekeeping values. * *p* < 0.01, ** *p* < 0.001, *** *p* < 0.0001.

**Figure 7 ijms-24-17084-f007:**
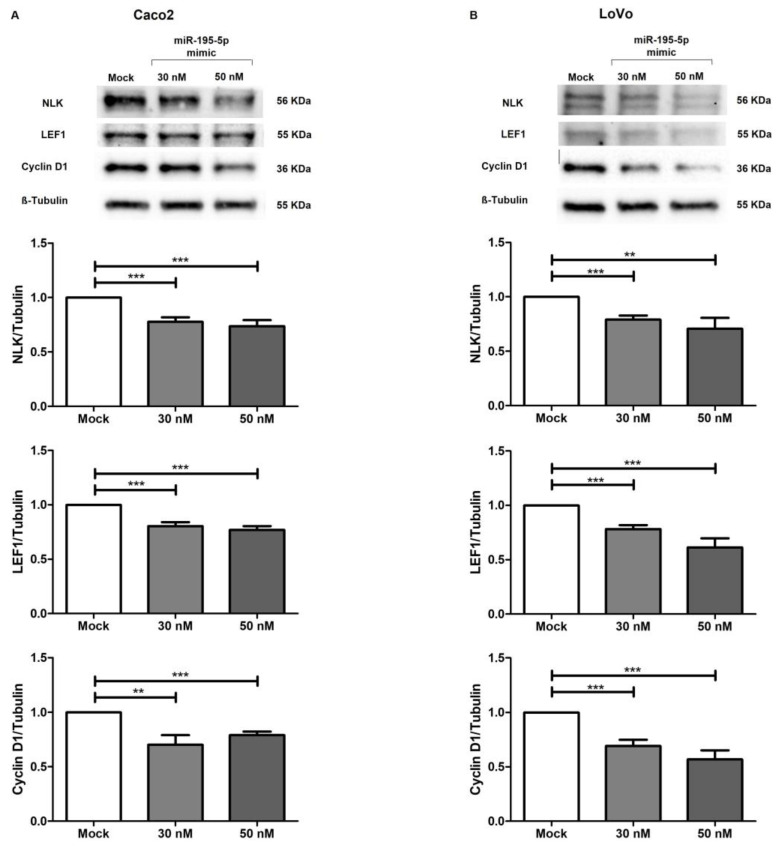
miR-195-5p regulated NLK, LEF1 and Cyclin D1 protein expression in all cell lines. WB analysis revealed that the gain of miR-195-5p in Caco2 (**A**) and LoVo (**B**) significantly decreased the expression levels of these key effectors of the Wnt pathway. Data obtained from three independent experiments (n = 3; mean ± SEM) were normalized to housekeeping values. ** *p* < 0.001, *** *p* < 0.0001.

**Figure 8 ijms-24-17084-f008:**
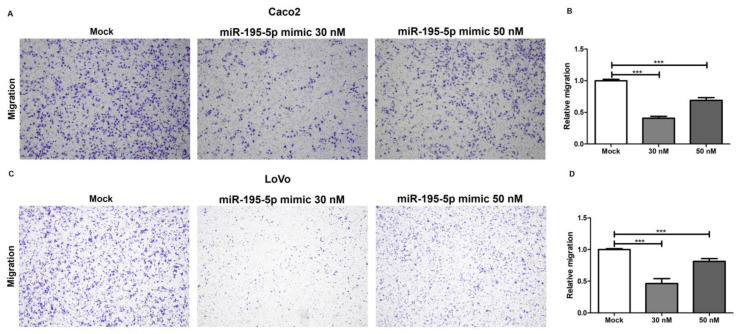
miR-195-5p inhibits the migration of CRC cell lines. The effect of miR-195-5p mimic on the migration of Caco2 (**A**,**B**) and LoVo cells (**C**,**D**) was evaluated by transwell assay. The cell migration ability was significantly reduced after miR-195-5p mimic transfection. Representative images of migrating cells with the corresponding quantitative analysis are shown. The number of migrated cells was counted in five randomly selected microscopic fields. Magnification 4×. The data are reported as relative level of migration of at least three independent experiments (n = 3; mean ± SEM). *** *p* < 0.0001.

## Data Availability

Data are contained within the article and Appendix A.

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
