# Peer review of "miR-195-5p as Regulator of γ-Catenin and Desmosome Junctions in Colorectal Cancer"

_ijms, 2023, doi:10.3390/ijms242317084_

Round 1

Reviewer 1 Report

Comments and Suggestions for Authors

In the present study, authors have investigated the role of miR-195-5p on desmosomal junction regulation and stated that miR-195-5p indirectly regulates important desmosomal proteins DSG2 and DSC2 via JUP. They have also shown that miR-195-5p regulates critical Wnt pathway effectors. After carefully reviewing the manuscript, I find that it needs additional study. Below are my comments:

1.       Authors have analyzed the miR-195-5p and JUP expression levels in publicly available datasets, but they have examined the miR-195-5p levels in one dataset, whereas the JUP levels are analyzed in another. It will be more convincing if they can show the same results in the same publicly available dataset or if they have their patient samples.

2.       It is unclear on what basis authors have selected JUP as a target to investigate further because, in their target analysis, there are >5000 targets having scored better than the JUP score. The authors have used only one software to predict targets, and it would be great if they could show a similar finding with any other target prediction software. It will be more helpful for other readers and help the reproducibility of the data if they can describe the criteria for target selection in detail.

3.       Authors have described that miR-195-5p plays an essential role in regulating desmosomal proteins, but they haven’t investigated phenotype in this study. It would be great if they could show whether the expression of miR-195-5p has any role in migration, proliferation, or anchorage-independent growth.

4.       This study has no mechanistic investigation or discussion about how miR-195-5p regulates JUP and desmosomal protein. It will make the manuscript more impactful if mechanistic details are mentioned.

5.       In this manuscript, all the miR-195-5p experiments are done with mimic, and it will be more impactful if these findings can be shown with the stable miR-195-5p expressing clones in these cell lines.

Author Response

In the present study, authors have investigated the role of miR-195-5p on desmosomal junction regulation and stated that miR-195-5p indirectly regulates important desmosomal proteins DSG2 and DSC2 via JUP. They have also shown that miR-195-5p regulates critical Wnt pathway effectors. After carefully reviewing the manuscript, I find that it needs additional study. Below are my comments:

We thank the referee for the critical and helpful evaluation and for the opportunity given to us to revise it.

  1. Authors have analyzed the miR-195-5p and JUP expression levels in publicly available datasets, but they have examined the miR-195-5p levels in one dataset, whereas the JUP levels are analyzed in another. It will be more convincing if they can show the same results in the same publicly available dataset or if they have their patient samples.

We appreciate the reviewer for the valuable suggestion. The dataset that we have previous analyzed and reported in our manuscript did not include miRNAs and mRNAs expression profile in the same patients. Hence, we have analyzed a new dataset from GEO (REF 39) in order to analyze the JUP and miR-195-5p expression levels from the same cohort of patients. We confirmed the previous data on miR-195-5p and JUP expression. We have also performed a Pearson correlation analysis that has revealed an inverse correlation between miR-195-5p and JUP expression levels. We have added these data in “Results” section, paragraph 2.3.

Moreover, several study supporting and confirming our results. For miR-195-5p expression levels in CRC, a widely integrated analysis of 10 colorectal cancer microRNA datasets performed on GEO DataSets clearly identified miR-195-5p as one of the most downregulated miRNAs in CRC (Falzone L, Scola L, Zanghì A, Biondi A, Di Cataldo A, Libra M, Candido S. Integrated analysis of colorectal cancer microRNA datasets: identification of microRNAs associated with tumor development. Aging (Albany NY). 2018 May 18; 10:1000-1014. https://doi.org/10.18632/aging.101444).

Regarding JUP expression, several works reported an overexpression of JUP in CRC patients compared to normal tissue. (Nagel, J.M., et al. γ-Catenin is an independent prognostic marker in early stage colorectal cancer. Int J Colorectal Dis 25, 1301–1309 (2010). https://doi.org/10.1007/s00384-010-1046-y; Luque-García, J.L., et al (2010), Differential protein expression on the cell surface of colorectal cancer cells associated to tumor metastasis. Proteomics, 10: 940-952. https://doi.org/10.1002/pmic.200900441; Sethi MK, et al. Quantitative proteomic analysis of paired colorectal cancer and non-tumorigenic tissues reveals signature proteins and perturbed pathways involved in CRC progression and metastasis. J Proteomics. 2015 Aug 3;126:54-67. doi: 10.1016/j.jprot.2015.05.037.)

  1. It is unclear on what basis authors have selected JUP as a target to investigate further because, in their target analysis, there are >5000 targets having scored better than the JUP score. The authors have used only one software to predict targets, and it would be great if they could show a similar finding with any other target prediction software. It will be more helpful for other readers and help the reproducibility of the data if they can describe the criteria for target selection in detail.

miRabel is a web platform that provide an intrinsic miRNA-mRNA binding score and moreover coordinate and integrate several miRNAs prediction software (miRanda, Pita, TargetScan and SVmicrO). In this way, our prediction analysis revealed a binding site between JUP mRNA and miR-195-5p that is common in miRabel, Pita and miRanda database, guaranteeing the reproducibility of the data

Among all predicted target genes,  we chose to study the effect of JUP regulation by miR-195-5p since in our previous works (REF 35-36) we have in vitro and in vivo characterized the role of miR-195-5p in the regulation of Tight Junction. Hence, in this study, with the aim of gaining further insight into the biological function of miR-195-5p in cell junctions, we investigated the effect of miR-195-5p in the modulation of desmosome complex. As reported in the manuscript, the target analysis performed by miRabel tool suggests a high affinity between miR-195-5p and JUP, an essential desmosome component that strictly interact with desmosomal cadherins.

  1. Authors have described that miR-195-5p plays an essential role in regulating desmosomal proteins, but they haven’t investigated phenotype in this study. It would be great if they could show whether the expression of miR-195-5p has any role in migration, proliferation, or anchorage-independent growth.

We thank the reviewer for the suggestion.

In order to further investigate whether the expression of miR-195-5p regulates the desmosome function, we carried out transwell migration assay. The gain of miR-195-5p in Caco2 and LoVo cell lines strongly reduced the migration potential and activity of cells as compared with mock control. These data demonstrated that the transfection with miR-195-5p mimic at 30 nM and 50 nM concentration reduced the migration ability of colon cancer cells in vitro.

We have added these data in “Results” section, paragraph 2.8.

  1. This study has no mechanistic investigation or discussion about how miR-195-5p regulates JUP and desmosomal protein. It will make the manuscript more impactful if mechanistic details are mentioned.

In the “Introduction” and “Discussion” sections we have argued the knowledge about the essential role of JUP in desmosmal complex. It is widely recognized that JUP is a major protein in cell-cell adhesion at the desmosomes, where it also strictly links transmembrane cadherins (desmogleins and desmocollins) to intermediate filaments of the actin cytoskeleton.(REF 7). DSG2 and DSC2 are two colonic specific isoforms of desmosomal cadherins. It has been demonstrated that DSG2 expression is low in colon cancer and correlates with poor survival (Yang, T., Gu, X., Jia, L. et al. DSG2 expression is low in colon cancer and correlates with poor survival. BMC Gastroenterol 21, 7 (2021). https://doi.org/10.1186/s12876-020-01588-2) and the loss of Dsc2 contributes to the progression of colorectal cancer cells (Kolegraff K, Nava P, Helms MN, Parkos CA, Nusrat A. Loss of desmocollin-2 confers a tumorigenic phenotype to colonic epithelial cells through activation of Akt/β-catenin signaling. Mol Biol Cell. 2011 Apr 15;22(8):1121-34. doi: 10.1091/mbc.E10-10-0845. Epub 2011 Feb 16. PMID: 21325624; PMCID: PMC3078068.)

We hypothesize that the mimic transfection could lead to an indirect regulation of DSG2 and DSC2 by miR-195-5p. Indeed, bioinformatically, we didn’t identify any miR-195-5p binding sites in the regulatory regions of DSG2 and DSC2 mRNA sequence, so miR-195-5p couldn’t directly bind mRNA sequence of DSG2 and DSC2 inhibiting its translation. Thus, our hypothesis was that the observed modulation of the desmosomal cadherins at protein level after transfection could be the result of JUP downregulation. Specifically, we suppose that miR-195-5p directly regulate JUP that in turn modulates DSG2 and DSC2 protein expression. In this way, miR-195-5p reduce the JUP dependent signaling and indirectly enhance the strength of cell-cell adhesion, upregulating the expression of DSG2 and DSC2.

In addition, we have further investigated the effects of miR-195-5p in the regulation of desmosome junctions performing a functional biological assay. The transwell assay revelead that the increase of the intracellular levels of miR-195-5p significantly inhibits the migration ability of CRC cell lines. These finding are strongly in accordance with the results obtained by Western blot analysis that have shown an indirect uperegulation of desmosomal cadherins by miR-195-5p suggesting the protective role of miR-195-5p in the establishment of cell-cell adhesion.

  1. In this manuscript, all the miR-195-5p experiments are done with mimic, and it will be more impactful if these findings can be shown with the stable miR-195-5p expressing clones in these cell lines.

We agree with the reviewer comments, since it could be very impactful to show our findings with the stable miR-195-5p expressing clones. Unfortunately, we are unable to perform these assays since we have not got the reagents in the lab and due to the 10-day revision interval, we could not acquire the assay and carry out again all experiments with the suggested molecules. Based on our experience is that not less than 3-4 months are required to obtain all results. However, future studies will be conducted according to the reviewer suggestion’s.

Reviewer 2 Report

Comments and Suggestions for Authors

The manuscript investigates the role of miRNA 195-5p on the regulation of the desmosome functions in colon cancer in vitro. The following aspects need to be addressed before publication:

  1. The most critical aspect is the lack of evidence supporting the participation of JUP in the regulation of the desmosomes and the WNT pathway in response to changes in the levels of miRNA 195-5p. The current evidence does not rule out a JUP-independent effect of this miRNA. Models with lack of function of JUP would be an alternative to prove participation of this protein. 
  2. Section 4.3. Please, provide a reference supporting the use of miRNA-26a-5p as reference in the real time PCR measurements. 
  3. Sections 4.4. and 4.5. Please, provide the dilutions of all the primary antibodies. 
  4. Figures 3 and 4. Please, indicate the number of replicates (n) in the figure legend. 
  5. Figure 4. Scale bars need to be added.
Comments on the Quality of English Language

Moderate English Editing required

Author Response

The manuscript investigates the role of miRNA 195-5p on the regulation of the desmosome functions in colon cancer in vitro. The following aspects need to be addressed before publication:

We thank the referee for the critical and helpful evaluation and for the opportunity given to us to revise it.

  1. The most critical aspect is the lack of evidence supporting the participation of JUP in the regulation of the desmosomes and the WNT pathway in response to changes in the levels of miRNA 195-5p. The current evidence does not rule out a JUP-independent effect of this miRNA. Models with lack of function of JUP would be an alternative to prove participation of this protein.

We appreciate the reviewer for the valuable suggestion. Unfortunately, we haven’t all the required reagents and the time of revision (10 days) was not sufficient to buy all and to perform the suggested experiments. We know that this could be a limitation of this work, however, as reported in Discussion section, this is a preliminary study with the initial exploration of the effect of miR-195-5p on desmosome junction regulation. We recognize that future studies will be needed to clearly validate the positive effectiveness of miR-195-5p in JUP-dependent CRC progression.

However, it was widely demonstrated that JUP has been shown to be crucially involved in the formation and maintenance of desmosomes anchoring intermediate-sized filaments by its interaction with the desmosomal cadherins, desmoglein (Dsg), and desmocollin (Dsc) and moreover acts as a signaling hub in Wnt pathway. (REF 7-14). Our hypothesis was that the observed modulation of the desmosomal cadherins at protein level after transfection could be the result of JUP downregulation. Specifically, we suppose that miR-195-5p directly regulate JUP that, in turn,modulates DSG2 and DSC2 protein expression. Moreover, we hypothesized that regulating JUP expression levels by the gain of function of miR-195-5p mimic, it was able in turn to indirect modulate the expression of key components of Wnt pathway, such as NLK, LEF1 and Cyclin D1.

In addition, in the present version of manuscript, we have further investigated the effects of miR-195-5p in the regulation of desmosome junctions performing a functional biological assay. The transwell assay revelead that the increase of the intracellular levels of miR-195-5p significantly inhibits the migration ability of CRC cell lines. These finding are strongly in accordance with the results obtained by Western blot analysis that have shown an indirect uperegulation of desmosomal cadherins by miR-195-5p suggesting the protective role of miR-195-5p in the establishment of cell-cell adhesion.

  1. Section 4.3. Please, provide a reference supporting the use of miRNA-26a-5p as reference in the real time PCR measurements.

The TaqMan Advanced miRNA Assays used for real time PCR measurements do not detect snRNAs or snoRNAs. According to manufacturer’s protocol, from a list of endogenous controls provided by Thermo Fisher Scientific, we choose miRNA-26a-5p as reference since it was one of the most stable miRNAs in CRC (Chang, KH et al. MicroRNA expression profiling to identify and validate reference genes for relative quantification in colorectal cancer. BMC Cancer 10, 173 (2010). https://doi.org/10.1186/1471-2407-10-173).

  1. Sections 4.4. and 4.5. Please, provide the dilutions of all the primary antibodies.

According to reviewer’s suggestion, we have added the dilutions of all primary antibodies in “Sections 4.4. and 4.5”.

  1. Figures 3 and 4. Please, indicate the number of replicates (n) in the figure legend.

As suggested by referee, we have added the number of replicates (n) in all figure legends.

  1. Figure 4. Scale bars need to be added.

We recognized that the scale bars in Figure 4 (now Figure 5) are not clearly visible, but they derived from the merged images for DAPI and TRITC channels. Unfortunately, we are unable to modify them and we added the information on images acquisition and scale bar in Figure 5 legend.

Round 2

Reviewer 1 Report

Comments and Suggestions for Authors

I would like to appreciate the authors for addressing my concerns regarding this manuscript and I am satisfied with the revised manuscript.

Author Response

We thank the reviewer for his/her positive response.

Reviewer 2 Report

Comments and Suggestions for Authors

The authors have addressed most of my concerns and provided an improved version. Noteworthy, evidence confirming the causal association between the regulation of JUP by the miR-195-5p and all the downstream effects (desmosomes, Wnt, cell migration) is still missing. However, the authors have provided a significant volume of data which may suggest that association. The following (minor) aspects should be checked before publication:

1- Line 131. It is not clear, which statistic analysis was performed. If a Pearson correlation analysis was performed, then a negative correlation coefficient (r) should be reported. 

2- Section 2.7. Since the confirming evidence for the participation of is still missing, the mention to JUP should be removed from the title of the section. 

3- There is a typo in line 225. 

4- Since the final confirming evidence of JUP participation is missing, the last sentence of the conclusions should be softened. 

Comments on the Quality of English Language

Minor editing required. 

Author Response

The authors have addressed most of my concerns and provided an improved version. Noteworthy, evidence confirming the causal association between the regulation of JUP by the miR-195-5p and all the downstream effects (desmosomes, Wnt, cell migration) is still missing. However, the authors have provided a significant volume of data which may suggest that association. The following (minor) aspects should be checked before publication:

We thank the Reviewer for the critical and helpful evaluation of our manuscript and for the opportunity given to us to improve it.

1- Line 131. It is not clear, which statistic analysis was performed. If a Pearson correlation analysis was performed, then a negative correlation coefficient (r) should be reported.

As reported in Results and Materials and Methods sections, we performed a Pearson correlation test to correlate mRNA and miRNA expression data. As suggested by reviewer, we have added the coefficient r in section 2.3 and Figure 3 legend.

2- Section 2.7. Since the confirming evidence for the participation of is still missing, the mention to JUP should be removed from the title of the section.

As suggested by reviewer, we changed the title of section 2.7

3- There is a typo in line 225.

We corrected it.

4- Since the final confirming evidence of JUP participation is missing, the last sentence of the conclusions should be softened.

We modified the last sentence of Conclusion section accordingly
